

# Automation and Heat Transfer Characterization of Immersion Mode Spectroscopy for Analysis of Ice Nucleating Particles

Charlotte M. Beall[1], M. Dale Stokes[1], Thomas C. Hill[2], Paul J. DeMott[2], Jesse T. DeWald[3], Kimberly A. Prather[1,4]

[1]Scripps Institution of Oceanography, University of California San Diego, La Jolla, 92037, USA
[2]Department of Atmospheric Science, Colorado State University, Fort Collins, 80523, USA
[3]Jacobs School of Engineering, University of California San Diego, La Jolla 92093, USA
[4]Department of Chemistry, University of California San Diego, La Jolla 92093, USA
*Correspondence to*: Charlotte M. Beall (cbeall@ucsd.edu)

**Abstract.** Ice nucleating particles (INPs) influence cloud properties and can affect the overall precipitation efficiency. Developing a parameterization of INPs in global climate models has proven challenging. More INP measurements, including studies of their spatial distribution, sources and sinks, and fundamental freezing mechanisms, must be conducted in order to further improve INP parameterizations. In this paper, an immersion mode INP measurement technique is modified and automated using a software-controlled, real-time image stream designed to leverage optical changes of water droplets to detect freezing events. For the first time, heat transfer properties of the INP measurement technique are characterized using a finite-element analysis based heat transfer simulation to improve accuracy of INP freezing temperature measurement. The heat transfer simulation is proposed as a tool that could be used to explain the sources of bias in temperature measurements in INP measurement techniques, and ultimately explain the observed discrepancies in measured INP freezing temperatures between different instruments. The simulation results show that a difference of +5.6 ℃ between the well base temperature and the headspace gas results in an up to 3 ℃ stratification of the aliquot, whereas a difference of +3.0 ℃ or less results in a thermally homogenous water volume within ±0.05 ℃. The results also show that there is a strong temperature gradient in the immediate vicinity of the aliquot, such that without careful placement of temperature probes, or characterization of heat transfer properties of the water and cooling environment, INP measurements can be biased toward colder temperatures. Using the modified immersion mode technique, the Automated Ice Spectrometer (AIS), measurements of the standard test dust illite NX are reported and compared against 5 other immersion mode droplet assay techniques featured in (Hiranuma et al. 2015) that used wet suspensions. AIS measurements of illite NX INP freezing temperatures compare reasonably with others, falling within the 5 ℃ spread in reported spectra. The AIS and its characterization of heat transfer properties allows higher confidence in accuracy of freezing temperature measurement, higher throughput of sample analysis, and enables disentanglement of the effects of heat transfer rates on sample volumes from time dependence of ice nucleation.

# 1 Introduction

## 1.1 Background



Ice nucleating particles (INPs) induce freezing of cloud droplets at temperatures above their homogeneous freezing-point (~-38 °C), and at a relative humidity (RH) below the homogeneous freezing RH of aqueous solution droplets at lower temperatures, influencing cold cloud lifetime, phase, as well as their optical and microphysical properties. INPs are comprised of a diverse population of particles, some species of which have complex sources and sinks; developing a

parameterization of INPs in global climate models (GCMs) that results in a credible representation of global cloud coverage and the radiative balance remains a challenge (DeMott et al., 2010; Seinfeld et al., 2016; Burrows et al., 2013). *In situ* observations to close critical knowledge gaps such as the vertical distribution of INPs in the air column, the complex sources and sinks of biological INPs, and INP influence on cloud microphysics are identified as a high priority for the improvement of INP representation in GCMs (Seinfeld et al., 2016; Burrows et al., 2013). One of the largest biases in shortwave

reflectivity exists over the Southern Ocean, and this bias may be influenced by poor representation of INPs over primarily oceanic regions (Trenberth and Fasullo, 2010; DeMott et al., 2010). Measurements of INP number concentrations, particularly in remote ocean regions are needed to help develop parameterizations of ice nucleation for use in cloud resolving models and GCMs. To further improve the parameterization of INPs, both field and laboratory measurements are needed identify drivers of ice nucleation in clouds. Accurately defining the activation temperature of INPs is critical to

understanding the influence of INPs on clouds and improving representation of INPs in GCMs because INP freezing temperatures influence cloud phase and lifetime in mixed-phase clouds, or the supersaturation or temperature conditions in which ice clouds can form (DeMott et al., 2003, Cziczo et al., 2013). INP concentrations applied in cloud and climate models must be accurate to within a factor of 10 to avoid biases that lead to significant differences in cloud radiative and microphysical properties (Phillips et al., 2003).

Several instruments and techniques exist, utilizing both online (real-time) and offline (processed post-collection) approaches, for the measurement of INP number concentration and activation temperature across the range of ice nucleation mechanisms. Ice nucleation mechanisms include deposition nucleation, immersion, contact and condensation freezing. However, some simulations find immersion freezing is the dominant ice nucleation mechanism globally from 1000 to 200 hPa (Hoose et al., 2010) hence, most INP measurement techniques target immersion mode freezing. In (Hiranuma et al., 2015), 17 online and

offline immersion mode instruments were compared using illite NX as the dust standard. The major differences between the 17 instruments studied are described in detail therein, however in brief, all of the instruments fall into one of two categories: droplet assay techniques, in which INPs are immersed in water and distributed among an array of pico to microliter scale droplets on a substrate and then cooled until frozen, or chamber techniques, in which droplets are passed through a temperature and humidity controlled chamber, where the freezing of droplets and their associated size change is detected

with optical particle counters. Each of these techniques pose significant INP measurement challenges due to the rarity of INPs, which represent 1 in $10^6$ or fewer of total aerosol particles (Rogers et al., 1998), and mitigation requires large air sample volumes which both limits the temporal sampling resolution, and increases the chance of contamination, which can overwhelm subtle INP signals in the data. Making INP freezing temperature measurements can also present challenges, because sample droplets or crystals cannot be directly probed with thermal sensors throughout the cooling process without



altering the fundamental shape or content of the droplet, and most thermal probes are not small enough to access nano to microliter sized droplets.

In this paper, an offline freezing assay technique for measurement of immersion mode INPs (Hill et al., 2016; Hiranuma et al., 2015) is automated using a software-controlled real-time image stream designed to leverage optical changes of water

volume arrays to detect freezing events. The offline freezing assay is an immersion mode technique that is similar to the immersion mode droplet assay, with a difference in the type of substrate used. In both techniques, multiple water volumes are supported on a substrate which is cooled until the water volumes are frozen, and concentrations of INPs as a function of freezing temperature are calculated from fractions of unfrozen droplets per temperature (see Sec. 2.1). In droplet assays, water volumes are distributed on a cold-stage as droplets during measurements. However, in the freezing assay, small

aliquots of water, typically around 50 $\mu L$ each, are distributed in 1.2 $mL$ wells within disposable polypropylene trays. The trays are mounted in aluminum blocks that are cooled during measurements (see Sec. 2.2). Albeit with significant loss of time resolution, droplet or freezing assays provide an offline alternative for INP measurement with fewer aerosol size limitations than online chamber techniques. For regular sampling on any surface site, INP samples may be collected on open-face filters, which reduce sample inlet particle size biases and particle losses.

There are ten current instruments for measuring immersion mode INP concentrations using picoliter to nanoliter droplet or liquid volume arrays on or within cooled surfaces: the Leeds Nucleation by Immersed Particles instrument (NIPI) (Whale et al., 2015), the Bielefeld Ice Nucleation ARraY (BINARY) (Budke and Koop, 2015), the North Carolina State Cold Stage (NC State-CS) (Wright et al., 2013), the CU-RMCS (Baustian et al., 2010; Wise et al. 2010), the Frankfurt Ice Nuclei Deposition FreezinG Experiment (FRIDGE) (Klein et al., 2010), the Colorado State University Ice Spectrometer (CSU-IS)

(Hiranuma et al. 2015, SI), the LED-based Ice Nucleation Detection Apparatus (LINDA) (Stopelli et al., 2014), the Cryogenic Refrigerator Applied to Freezing Test (CRAFT) (Tobo, 2016), WISDOM (WeIzmann **S**upercooled Droplets Observation on Microarray) (Y. Rudich, personal communication), and the MicroOrifice Uniform Deposit Impactor-Droplet Freezing Technique (MOUDI-DFT) (Mason et al., 2015). Differences between the techniques include: a variety of strategies to minimize the Wegener-Bergeron-Findeisen process, in which frozen droplets near liquid droplets grow faster by taking up

water vapour, the degree to which RH or evaporation is controlled, number and size of droplets (or samples) accommodated, measurable freezing temperature range, and how freezing events are detected. Almost all of the ten techniques, with the exception of the CSU-IS, use a camera to image the droplets. The NIPI, FRIDGE, and NC-State CS save images at a frequency on the order of 1 image per second, and post-process a stream of images of the droplets with varying levels of automation in the determination of freezing events. BINARY and LINDA report the use of an algorithm similar to the one

described here (see Sec. 2.2) in which changes in the 8-bit mean grey value of a monochrome image, or the intensity of LED light transmitted, respectively, are used to detect the droplet's phase change. The heat transfer properties of the instrument are also characterized through a finite element analysis heat transfer simulation to evaluate the homogeneity of INP sample temperatures and identify optimal locations for the thermal probes. Finally, the standard test dust used in (Hiranuma et al.,

2015) was tested using the instrument, and was compared against the 6 other droplet array immersion mode INP measurement techniques that reported wet suspension measurements of illite NX.

## 2 Automation of Immersion Mode Ice Spectroscopy

### 2.1 Theory of Operation

Immersion mode ice spectroscopy measures INP concentrations at specific temperatures of a liquid sample. INP measurements of air samples are made by collecting particles on a filter (or via impinging particles into liquid), immersing the filter in ultrapure water, and shaking particles off of the filter by hand or via an automated rotator (DeMott et al., 2016). The liquid sample is then distributed in microliter aliquots into a clean 96-well disposable polypropylene sample tray. An equal number and volume of aliquots of ultrapure water accompany each sample in the disposable tray as control for contamination from the loading and/or ultrapure water. The sample trays are then inserted into an aluminium block that is cooled until the samples are frozen. The homogenous freezing point of water is -38 ˚C, but the 96-well sample tray surface induces freezing at higher temperatures, typically starting at -25 to -27 ˚C, which limits the lower temperatures for which INP number concentrations may be assessed. Cumulative INP number concentrations per temperature per volume are calculated using the fraction of unfrozen wells $f$ per given temperature interval:

$$INP = \frac{-ln(f)}{V}$$

Eq. (1)

where V is the volume of the sample in each well (Vali, 1971). The fraction of unfrozen wells $f$ is adjusted for contamination by subtracting the number of frozen ultrapure water wells per temperature interval from both the total number of unfrozen wells and total wells of the sample.

### 2.2 Physical Design and Automation of the Ice Spectrometer

In previous Ice Spectrometer INP studies, observation of each well freezing event was conducted manually by an operator, which limited the number of samples and wells per sample that could be processed, and required a cooling rate slow enough to accurately observe and manually record each freezing event. In order to increase sample throughput and improve accuracy of INP freezing temperature measurement, the immersion mode ice spectrometer (Hill et al., 2014; Hiranuma, et al. 2015) was redesigned to increase sample cooling rates (see Figs. 1 and 2) and automated using a software-controlled camera that monitors changes in optical properties of water droplets during freezing. In this paper, the new instrument's max average cooling rate of -0.86 °C min$^{-1}$, as measured in the coolant bath from room temperature to -27 °C, was used for all measurements and simulations. For the same temperature range, the average cooling rate as measured at the base of the well





is the same. We do not investigate the role of cooling rate on freezing, known to influence freezing activation spectra to a much smaller extent than temperature alone (Vali, 2014).

In the new instrument, the Automated Ice Spectrometer (AIS), two aluminium well blocks are fixed inside the coolant bath cavity of a Fisher™ Isotemp™ Refrigerated Bath Circulator and fitted with a sealed splash guard to prevent contamination

of the well region by contact with the coolant. Each of the two aluminium blocks has a machined indentation cavity in which the 96-well disposable sample tray is tightly fitted. A plexiglass lid caps the well region to insulate and isolate the air above the wells from room temperature air. A 1.8 m long, 0.64 cm diameter coiled copper tube, connected to an external dry nitrogen supply, lies in the coolant bath beneath the well block in order to cool nitrogen gas that is pumped over the well region at 0.25 Lpm. The cold nitrogen gas purges room temperature air from of the well region to decrease stratification of

temperature within the sample volumes. A flow rate of 0.25 Lpm was chosen because it was found empirically to most effectively cool the air above the well region. The nitrogen gas enters the well region slightly warmer than the chilled bath temperature (about +2-5 °C, see Fig. 5) because the gas flows through approximately 15 cm of rubber tubing exposed to the ambient room temperature before being injected beneath the plexiglass cover. At flow rates less than 0.25 Lpm, room temperature air leaks into the well region, but at significantly higher flow rates, the fast flowing nitrogen gas lifts the acrylic

plate causing additional leakage.

A 0.5 Megapixel monochrome camera (Point Grey Blackfly 0.5MP Mono GigE POE) is used to image the wells throughout the cooling process. As depicted in Fig. 3, the camera is controlled with custom National Instruments, LabView software which allows the user to adjust imaging parameters including brightness, exposure, gain, and rotation via a graphical user interface control panel shown in Fig. 4. The refrigerated bath circulator is also controlled by the software and allows the user

to either ramp the temperature of the coolant from room temperature to the input target temperature at a constant rate or to "stair-step" the coolant bath temperature at adjustable, incremented time and temperature steps (with the tolerance of the bath circulator thermostat, the starting and stopping temperature as additional input options).

As shown in Figs. 1 and 2, the camera is fixed above the well region at the top of a plastic housing fabricated from white cast acrylic sheet (0.64 cm thick). An adjustable cradle holds the camera and allows aligning of the camera lens (2.8-12 mm

Focal Length, Varifocal Video Lens, Edmund Optics) over the centre of the well block. Also fixed within the white housing are two white LED backlights (Edmund Optics), one on either side of the well region, which together provide a stable lighting environment for imaging of the wells and the liquid samples. Once the camera is aligned using the adjustable cradle, the video image is live-streamed via the control software so that two 8x12 grids of 15x15 pixel squares are aligned over all 192 wells. Each 15x15 pixel box corresponds to an individual sample well and the mean intensity of light reflected from

each well is recorded.

When droplets freeze, the intensity of the light reflected back to the camera decreases due to the dark background of the inner well block. As in the flowchart depicted in Fig. 3, at each new time step $t_i$, if the difference between the mean intensity $I$ of the well at $t_i$ and the mean intensity of the well at $t_{i-1}$ is greater than the set pixel change threshold $\eta$ such that





$I(t_i) - I(t_{i-1}) > \eta$, a freezing event is detected and time, freezing temperature, and location of the well is recorded. The exposure, gain, and pixel change threshold $\eta$ can be adjusted in the control panel to increase the signal to noise ratio by emphasizing the decrease in mean intensity due to freezing, and minimizing the background variation in mean intensity due any oscillation of the chiller unit when the coolant circulator is running. Temperature measurements are made with a

thermistor imbedded at the base of a well in the sample tray after threading the sensor leads through a small hole drilled in the aluminium block.

## 3 Simulation of Heat Transfer for Immersion Mode Ice Spectroscopy
### 3.1 Model design

In order to accurately measure the freezing temperature of INPs in immersion mode spectroscopy, the temperature of each well must be quantified, and the temperature of the sample throughout the volume itself must be homogenous (unstratified). Placing thermistors directly in the sample volume would be ineffective for several reasons including: (1) the probe itself disrupts the structure of the surface of the droplet and could provide a surface for nucleation, (2) heat conducts through the

probe into the sample volume and (3) probes can introduce contamination. Also, if a probe is placed in a sacrificial sample well, once the well freezes, latent heat is released and because the thermal properties of ice are different from those of water, the temperature of the frozen well may not be representative of the supercooled liquid wells. Thus, the probe must be placed outside the well volume but in a region of the well block that is thermally homogenous with the sample. Alternatively, if the heat transfer characteristics of the system are resolved, the thermal probe could be placed anywhere in the block where the

offset in temperature between the probe's location and the sample well volume is quantified. The sample volume itself must be thermally homogenous because if the sample volumes were stratified, a freezing event could be triggered in any of the stratified well layers depending on its temperature and the buoyancy of the ice nucleating entity.
In order to address the thermal properties of the aluminium block and well-plate system, a finite element analysis-based heat transfer simulation was developed using the 3D Design Software SolidWorks to investigate the homogeneity of temperature

within the 50 microliter sample volumes throughout the cooling process, and to determine the optimal placement and number of thermistors needed to resolve the temperature of each well. As shown in Fig. 6, a 3-D model of the AIS was designed using the dimensions and material properties of the actual instrument components. In finite element analysis heat transfer simulations, a mesh is applied to the modelled object such that, with a given initial temperature and/or heat source at the boundaries, rates of heat transfer and temperature are computed iteratively until solutions converge on the user-defined

mesh. Meshing becomes more computationally expensive over curved or complex surfaces, and because the AIS well blocks contain 192 wells each with a curved inner surface, a cut across the upper-left corner shown in Fig. 6a was made in the 3-D model to reduce computation time. In Fig. 6a, the two aluminium 96-well blocks are shown with the PVC splash guard, and the black line in the upper-left corner represents the modelled cut in the well block. Figures 6b and 6c show a close-up of the corner featured in the model, including the aluminium well block, the polypropylene sample tray, and a 50





microliter sample of water. A pocket of gas between the sample tray and the well block is also modelled due to the slightly imperfect fit of the tray to the well block in the actual instrument.  The simplifying modelling cut was justified by making measurements of the horizontal distribution of temperature through the two boundaries of the well region: the nitrogen gas above the well region and the coolant bath, which by design maintains a homogenous temperature throughout the coolant

volume.  The homogeneity of temperature in the coolant bath was verified using a calibrated thermometer (Checktemp Pocket Thermometer Hanna Instruments, accuracy ± 0.3 °C from -20 to 90 °C). To investigate the horizontal distribution of the temperature of gas across the surface of the well block, 4 thermistors were placed in the 5 cm headspace between the well block surface and the plexiglass lid during repeated cooling processes, and the thermistor temperature was monitored while systematically moving the thermistors through the headspace. The temperature of the nitrogen gas in the headspace was

found to be homogenous across the plate within ± 0.3°C (within the error of the calibrated temperature probe).
The horizontal gradient of temperature is constrained by the homogenous temperature across bottom surface of the well block and a temperature difference of max ± 0.3 °C across the top surface. The vertical gradient of temperature through the well block, disposable sample tray, and sample volume is not practically measurable and requires resolution through heat transfer simulations in order to determine where probes should be placed to measure temperature of the wells. The larger

hole on the left side of the sample well in Figs. 6a and 6b is where the thermal probes were placed in the original AIS design. The mesh used is shown in Fig. 6c and was applied using the SolidWorks standard mesh solver.  It is composed of discrete, tetrahedral elements that are connected at the three nodes such that they converge through all components in the modelled system.  Generally, an aspect ratio around 1 for each element is ideal, and for the mesh applied in the heat transfer simulations, 96.1% of the mesh elements with an aspect ratio of less than 3, and 0.07% of elements with an aspect ratio

greater than 10. Four Jacobian points, or nodes at the midpoint of element sides, were applied to each element to align with curvature more effectively with linear elements, and the mesh took 7 s to converge.

**3.2 Set-up of the heat transfer simulation**

The nitrogen and coolant fluid in thermal contact with the sample volumes and well block, respectively, form the thermal boundaries of the simulation. Thus, to quantify the boundary conditions for the heat transfer simulation, temperature measurements were made of the gas temperature above the sample volumes and the coolant temperature during a ramp cooling process, in which the refrigerated bath circulator ran from room temperature to -27 °C at an average cooling rate of -0.86 °C per minute (see Fig. 5). In addition, a hole was drilled into the aluminium block so that a thermistor could be placed

directly underneath a sample well.
Once the thermistor was placed in the block, the hole was sealed with acrylic caulk to prevent coolant fluid from entering the well region, and heat sink compound was applied to the thermistor so that it was in thermal contact with the aluminium block and the disposable sample tray. In Fig. 5, the temperature at three locations within the AIS are shown after measurement throughout a "ramp" cooling process from 0 °C to -35°C: 1) the coolant in contact with the bottom surface of the well block,


2) the gas above the sample volume, or headspace gas, and 3) directly below the sample well. The measurements of temperature of the gas above the sample volume and coolant over 1320 s of cooling are applied as boundary conditions in the heat transfer simulation. The larger plot in Fig. 5 shows the warm temperature offset of the headspace gas from the measured temperature at the well base, and the inset plot shows temperature changes in time, at the three locations over the

ramp cooling cycle. The headspace gas and coolant temperature data applied as boundary conditions in the simulation are shown highlighted in yellow in the inset. Figure 5 shows that the air above the well region is warmest, a maximum of +5.28 °C warmer than the well base, despite the chilled nitrogen pumped over the well region because the system is imperfectly insulated from the room temperature environment and because there is a slight warming of the gas before it enters the headspace (as described in Sec. 2.2). An acrylic plate covers the wells as shown in Fig. 1, but the system is not thermally

isolated from the environment.

Figure 6c shows each of the components considered in the model: the aluminium well block, the disposable sample tray, the gas pocket in the gap between bottom of the sample tray and the well block, and the 50 microliter sample water volume. The coolant and the headspace gas were considered as variable thermal loads to the system rather than included as components. Two types of heat transfer were considered during the model analysis: conductive and convective. All of the components

shown in Fig. 6d are considered to be bonded, or treated as if heat transfer by conduction occurs in a continuous manner. Heat transfer by conduction is computed at each element of the mesh by the following equation:

$$Q_{conduction} = kA(T_{hot}\text{-}T_{cold})$$   Eq. (2)

where $Q_{conduction}$ is the rate of heat transfer in Watts, $k$ is thermal conductivity of the component, $A$ is the heat transfer area defined by the mesh, and $(T_{hot}\text{-}T_{cold})$ is the temperature difference between the two mesh elements considered. Thermal conductivity, $k$, is determined by the material of the component. Values of $k$ used in the simulation are shown in Table 1. At all interfaces where the model is in contact with headspace gas, heat transfer by convection is considered. For heat transfer by convection, Eq. (3) is applied at each element:

$$Q_{convection} = hA(T_s\text{-}T_f)$$   Eq. (3)

where $Q_{convection}$ is rate of heat transfer from a body to a fluid in Watts, $h$ is the heat transfer coefficient in W/m²K, $(T_s\text{-}T_f)$ is the difference in temperature between the surface of the body and the fluid. $A$ is the same as above in Eq. (2). The

convection of both the gas and the water in the model was considered natural convection rather than forced. Typical ranges for the heat transfer coefficient $h$ for natural convection of air are 5-25 W/m²K (Yousef et al., 1982). The model output was insensitive to this range of coefficient variability, and a value of 5 W/m²K was used. The range of $h$ for natural convection of water however is much larger, from 2-3000 W/m²K (VDI-Gesellschaft Energietechnik, 2013), so $h$ was estimated by



approximating the wells as two vertical plates and calculating the Nusselt number, where $h = Nk/H$, where $N$ is the Nusselt number, and $H$ is the height of the plates. $N$ was calculated using Eq. (4) for laminar flow:

$$N = 0.68 + \frac{\left(0.670 Ra^{1/4}\right)}{\left(1+\left(\frac{0.492}{Pr}\right)^{9/16}\right)^{4/9}}$$

Eq. (4)

Ra and Pr are the Raleigh and Prandtl number, respectively, where

$$Ra = \frac{g\beta(T-T_\infty)D^3}{\upsilon} * Pr$$

Eq. (5)

and $Pr = \upsilon/\alpha$.

$\beta$ is the coefficient of thermal expansion, $T$ is the temperature of the water volume, $T_\infty$ is the temperature of the air at the surface of the water volume, $D$ is the diameter of the well as measured at the top of the well of the disposable sample tray, $\upsilon$ is dynamic viscosity, and $\alpha$ is the thermal diffusivity Since $\beta$, $\upsilon$, $\alpha$, and k are temperature dependent properties, and $h$ is of interest over the supercooled range from 0 to -25 °C, $N$ and $h$ were calculated at -5 °C, -15 °C, and -25 °C, using

corresponding values of $\beta$, $\upsilon$, $\alpha$, and k (Kell, 1975; Dehaoui et al., 2015; Benchikh et al., 1985; Biddle et al. 2013), which are shown in Table 2. Thus, $h$ was estimated to be 161, 191, and 202 W/m²K at -5 °C, -15 °C, and -25 °C, respectively. Within the range 161-202 W/m²K, the model was insensitive and a constant value of 191 W/m²K was used throughout the simulations.

The simulation was run over 1320 s with two different sets of boundary conditions. In the first, the gas above the well was

given a constant offset of +12 °C from the coolant bath temperature over the 1320s using the data from Fig. 5, and in the second the gas and coolant temperatures were applied directly from the data spanning the yellow bar in Fig. 5. The first condition has warmer headspace gas temperatures than those that were measured during the cooling process on the actual instrument in Fig. 5. This condition represents a doubling of the warming of the gas after it exits the coolant bath and before it enters the headspace (as described in Sec. 2.2), and could be considered representative of an instrument with inefficient

cooling of the gas above the well region. However, the difference in temperature between the well and headspace gas between the two cases is less than double due to the time delay of heat transfer through the block. For example, the maximum offset between the well and the warmer headspace gas in the second simulation (measured conditions) is +5.7 °C at 960 s, whereas in the first simulation, the maximum offset between the well and the warmer headspace gas is +7 °C at 1080 s.


**4 Results**

**4.1 Simulation results**



The results of the heat transfer simulation for the +12 °C gas temperature offset condition and the measured gas and coolant temperature conditions are shown in Figs. 7 and 8.  Fig. 7 shows a graphical time series of the heat transfer simulation with the +12 ℃ temperature offset between the chiller and the warmer gas above the well region.  The heat distribution is shown
in 12 time steps at 110s intervals over a 1320s simulation, with the coolant fluid cooling from -8 °C  to -25 °C over that period (i.e. -0.69 ℃ min$^{-1}$).  The average cooling rate for these selected temperatures is slower than the average cooling rate of -0.86 °C min$^{-1}$ measured for the room temperature to -27 ℃ range because the rate of cooling slows as the refrigerated cooling bath approaches its minimum temperature of -35 ℃.  At the top of Fig. 7, an isometric view of the well at 660 s is shown, and to the right is a detailed view of the well. The results show the stratification of temperature in the sample volume
itself, ranging from -5 °C at the skin of the sample volume, to -8 °C at the bottom of the sample volume. These results demonstrate that with a +12 °C offset between the well and the headspace gas, the temperature difference of +5.6 ℃ between the well and the headspace gas is too large to maintain homogenous temperature within the liquid sample volume which then becomes stratified.

Fig. 8 shows a graphical time series of the heat transfer simulation with the measured AIS headspace gas temperature and
coolant bath temperature conditions. With an offset between the base of the well and the headspace gas temperature of +3.0 °C, the temperature is homogenous throughout the liquid sample volume within ± 0.05 °C.

The results also show that the distribution of heat throughout the well block requires careful placement of the temperature probe such that the temperature of the probe location is accurately indicating the temperature of the sample volume. In each of the simulations, the sample water volume comprises the warmest body in the model assembly. Throughout the modelled
assembly, the temperature at the top of the gas pocket underneath the well of the polypropylene disposable tray was the closest with the sample water temperature, (within ±0.15 ℃).  Due to strong temperature gradients between the water sample and the immediately surrounding aluminium block, small variations in probe location can result in disproportionately large temperature offsets from the sample volume.   At 660 s in the second simulation, which applies the gas and coolant temperature conditions as measured on the AIS (Fig. 5), the temperature decreases ~ 1 °C from base of the well of the
polypropylene disposable tray through the gas pocket to the aluminium surface of the well block over a distance of 2.5 mm. This could be caused by the high specific heat of the water volume relative to the aluminium and the insulating thermal properties of the polypropylene tray could be responsible for the strong temperature gradient. In the current design of the AIS, the thermal probe is located in this gas pocket, and the simulation results suggest that at this location there could be up to a 1°C cold bias in the INP freezing temperature measurements. Thus, during ramping of the coolant bath from room
temperature to -27 °C at about -0.86 ℃ per minute, there is nowhere to place a probe in the aluminium block where the temperature perfectly matches that of the liquid sample volume (within << 1 °C). The offset in temperature between the probe and the sample temperature should be quantified so that recorded temperatures can be adjusted accordingly.  Ideally, any modifications made to the AIS system to fit a thermal probe, such as holes drilled in the aluminium or addition of heat sink compound should also be represented in the model when determining an offset. A slower or stair-step method of



cooling, where the temperature of the chiller is set and held for a set amount of time, would increase the region of thermal homogeneity in the well block but it would still be necessary to characterize the heat transfer properties of the system to determine how long the region takes to reach thermal equilibrium and where the probe should be optimally placed. Failure to resolve the heat transfer characteristics of an immersion mode system could result in either warmer or colder temperature

biases in measurements.

## 4.2 Automated Ice Spectrometer Performance: comparison with 5 other immersion mode ice nucleation measurement techniques

The accuracy of the Automated Ice Spectrometer (AIS) INP concentration measurements were evaluated using a standard, well-characterized test dust that has previously been used to compare immersion mode ice nucleation measurement techniques, illite NX (Arginotech, NX nanopowder) (Hiranuma et al., 2015). A suspension of dust and Milli-Q ultrapure water was prepared in a sterile 50 mL centrifuge tube (Corning) using a sample from the same batch of illite NX used in (Hiranuma et al., 2015) in a study of 17 immersion mode ice nucleation measurement techniques.  Dilutions of 1:25,000,

1:20,000 and 1:2,000 were used, corresponding to a range of $4.0 \times 10^{-5}$ to $5.0 \times 10^{-4}$ wt %.  For comparison, in (Hiranuma et al., 2015), droplet assays were intercompared using illite NX suspensions of varying dilutions within the range of $3.1 \times 10^{-6}$ wt % to 1.0 wt %. Higher concentrations of illite NX solution were not measured using the AIS because the automation software requires an optically clear solution to detect freezing events. 50 $\mu$L aliquots of the suspension were loaded into 24 wells of the disposable sample tray (Life Science Products™ 96 well PCR plates), and 24 adjacent wells were filled with 50

$\mu$L aliquots of Milli-Q water.  Prior to loading, the plexiglass lid was cleaned with an isopropyl alcohol based surface cleaner, rinsed three times with Milli-Q and dried with clean compressed air, and nitrogen was pumped over the well region at 0.25 Lpm for 20 minutes to purge the lines of any dust.  The loaded and covered sample was then cooled from room temperature to -27 ˚C  (with an average cooling rate of -0.86 ℃ min$^{-1}$), at which point the Milli-Q water had frozen in all wells.  The experiment was repeated 3 times.  Freezing events were detected using the automation software and the time of

freezing, well temperature and sample number recorded into an ASCII file for further analysis. Cumulative concentration of INPs per volume per 0.25 ℃ were calculated using Eq. (1) from (Vali, 1971) $-\ln{(f)}/V$, where f is the fraction of wells unfrozen, and V is the sample volume.  Although the placement of the temperature probe was as described in Sec. 4.1, no offset to the temperature was applied in the measurements.  In order to compare directly with (Hiranuma et al., 2015), cumulative concentrations of INPs were converted into a surface-site density, $n_{s,BET}$. The specifics of the parameterization

are in (Hiranuma et al., 2014), but briefly, the parameterizations are based on BET (Brunauer Emmet Teller) (Brunauer et al., 1938) $N_2$-adsorption-based specific surface area in which the particle surface area is measured based on the quantities of a variety of gases that form monolayers on the surface of the particle. The specific surface area ($SSA$) of the illite NX sample used in (Hiranuma, 2015) was 124 m$^2$/g and the mass concentration ($m$), of the three illite NX solutions processed in the AIS



ranged from 4.0 x 10$^{-5}$ to 5.0 x 10$^{-4}$ g/mL . Cumulative INPs per volume as measured by the AIS were divided by this mass concentration and converted to $n_{s,BET}$ by dividing by the specific site density as:

$$n_{s,BET} = \frac{\left(\frac{INPs}{mL}\right)}{SSA \cdot m}$$

Eq. (6)

In Fig. 9, the measured illite NX spectra are shown with the 6 of the 17 total ice nucleation measurement techniques from (Hiranuma et al., 2015). These were similar freezing or droplet assay techniques: the CSU-IS, the NIPI, FRIDGE (in immersion mode), NC-State CS, and the CU-RMCS. The 5 instruments made wet suspension-based measurements of illite NX in ultrapure water rather than dry-particle based measurements, and thus should more directly compare to those of the

AIS. The ice nucleation surface site density spectra of the 5 measurements fall within a range of about 5 °C, and the Automated Ice Spectrometer measurements compare favourably to the those of the other 5 techniques through its final temperature of -19 °C.  However, the AIS measurements fall on the warmer side of the temperature spectrum from -10 to -15 °C. Based on the results of the heat transfer simulations in Sec. 4.1, differences in the cooling process type (stair-step or ramp), location of temperature probe or method of freezing temperature measurement could have strong influences on

reported freezing temperatures.  These factors might account for some of the 8 °C  (or 5 °C for wet suspension droplet assay techniques) spread in spectra reported in (Hiranuma et al., 2015).

**5 Discussion**

The immersion mode ice spectrometer (original configuration in Hill et al., 2014, and latest design described in Hiranuma et

al. 2015) was modified to fit inside a refrigerated circulating coolant bath, and automated using a software controlled camera.  Older versions of the immersion mode ice spectrometer were designed with the aluminium well blocks external to the refrigerated circulator bath and the coolant fluid was pumped through heat exchange plates encasing the aluminium well blocks for cooling via external copper tube plumbing. The operator observed and recorded well freezing manually. Modifications to the instrument increased thermal homogeneity across the well block by immersing well blocks directly in

the coolant bath.  Automation enables more objective and instantaneous recording of well freezing events, and frees the operator from having constantly monitor sample processing.

The heat transfer properties of the AIS were characterized using finite element analysis heat transfer simulations, with measured temperatures of the well block headspace gas and the coolant bath applied as boundary conditions.  Heat transfer by conduction and convection were considered.  While temperature homogeneity across the well block horizontally was not

investigated through the simulation, the temperature across the block was constrained by measurements of the coolant fluid and gas above the well block to be consistent within ± 0.3 °C.



The results of the simulations showed that efficient cooling of the well block head space, with a maximum +3 ℃ offset between the base of the well and the headspace gas, or +6 ℃ between the coolant bath and the headspace gas, is necessary to ensure that the liquid sample volume is unstratified within ± 0.05 ℃, so that the well freezing temperature is representative of the population INPs in the well.  The results also demonstrate a strong temperature gradient from the sample volume to

the polypropylene and aluminium immediately surrounding the sample, of up to -1 ℃ in the 2.5mm gap. Thus the temperature measurement in the AIS is highly sensitive to the location of the thermal probe. In the simulation, the only region with a temperature consistent with the sample volume was the top of the gas pocket between the bottom of the polypropylene disposable tray and the aluminium block. However, a thermistor probe cannot physically fit in this small region, so INP freezing temperature measurements are likely biased by the thermistors contact with the aluminium block.

For other immersion mode droplet assay INP measurement techniques, variation in heat transfer properties and thermal probe placements may result in higher or lower accuracy of INP freezing temperature measurement, but the sensitivity of the temperature gradient within the droplet to the thermal heterogeneity of its cooling environment, as well as that of the temperature measurement to thermal probe placement motivates careful study of the effect of heat transfer properties of the various techniques.  The heat transfer simulations applied here could support investigations of bias in temperature

measurement for INP measurement techniques, and ultimately help decrease disparities between various instruments. INP concentrations applied in cloud and climate models must be accurate within an order of 10 to avoid propagation of error leading to significantly different cloud properties (Phillips et al., 2003), and as measurements typically show INP concentrations increasing with decreasing temperature in complex multi-exponential functions (Hiranuma et al., 2015), an 8 ℃ uncertainty in freezing temperature measurement could result in vast differences in model output. Heat transfer

simulations could prove particularly useful in studies of the role of varied cooling rates on assessment of ice nucleation activity in different devices due to the stochastic or time-dependent nature of droplet freezing at a given temperature. In such an investigation, it is important to separate the impact of time dependence of the ice nucleating entity from variations due to temperature gradients between the location of the thermal probe and the sample volume.

Fast cooling of samples (>1 ℃/min) has been discussed as a potential source of stratification of temperature between the

substrate and the droplets, or within the droplets, and conversely, that chilled nitrogen in the headspace might not be necessary to avoid stratification (Tobo, 2016).  However, the heat transfer simulation results below show that even with cooling rates below 1 ℃ min$^{-1}$, stratification within the sample volume can occur, and that the temperature of nitrogen gas in the headspace may play a significant role in controlling temperature stratification within the droplets.

The performance of the Automated Ice Spectrometer was evaluated using measurements of illite NX, a well characterized

test dust that has been used to intercompare 17 immersion mode INP measurement techniques. Three different dilutions of illite NX suspension were measured: $4.0 \times 10^{-5}$, $5.0 \times 10^{-5}$, and $5.0 \times 10^{-4}$ wt %.  These concentrations fall in the lower end of the range of suspension concentrations ($3.0 \times 10^{-6}$ to 1.0 wt %) measured by the 5 selected droplet assay INP measurement techniques in (Hiranuma, 2015) (see Fig. 9).  Measurements of specific site density compare well with the 5 droplet assay



techniques from the intercomparison study (Hiranuma et al., 2015), falling on the warmer side of the 5 ℃ spread in the

reported spectra from -10 to -15 ℃.

In summary, the Automated Ice Spectrometer:

1.    enables entirely autonomous measurement of INP concentrations

2.    can measure concentrations of INPs with activation temperatures in the range 0 to -25 ℃

3.    can process up to 7 samples per hour (including time for loading samples)

4.    has characterized heat transfer properties so that temperature offsets between temperature probes and the sample

        volume are identified

*Acknowledgements.*  We would like to thank Joe Mayer for his invaluable help in constructing the AIS housing and camera
mount. Funding to support this research was provided by the Center for Aerosol Impacts on Climate and Environment
(CAICE), an NSF Center for Chemical Innovation (CHE-1305427).

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

**Table 1: Simulation components and properties used in heat transfer simulation**

| Components | Material | $k$ (W/mK) | $h$ (W/m$^2$K) |
|---|---|---|---|
| Well block | Aluminum 1060 Alloy | 200 | 2.0 |
| Disposable sample tray | Polypropylene | 0.03 | 5 |
| Gas pocket | Air | 0.12 | n/a |
| Liquid INP sample | Water | 0.5 | 191.0 |

**Table 2: Constants used in calculation of $h$, heat transfer coefficient for water in natural convection from -5 to -30 ºC**

| Water Temperature T (ºC) | Gas Temperature $T_\infty$(ºC) | $\beta^\dagger$ (K$^{-1}$) $\times 10^{-6}$ | $\upsilon^{\dagger\dagger}$ (m$^2$/s) $\times 10^{-6}$ | $\alpha^*$ (m$^2$/s) $\times 10^{-7}$ | $k^{**}$ (W/mK) | $h$ (W/m$^2$K) |
|---|---|---|---|---|---|---|
| -5.0 | -1.6 | -168.6 | 2.0026 | 1.30 | 0.520 | 160.8 |
| -15.0 | -10.3 | -450.3 | 3.0707 | 1.20 | 0.500 | 191.0 |
| -30.0 | -23.4 | -1400.0 | 7.9703 | 1.05 | 0.450 | 201.6 |

† Kell, 1975
†† Dehaoui, 2015
* Benchikh, 1985



** Biddle, 2015

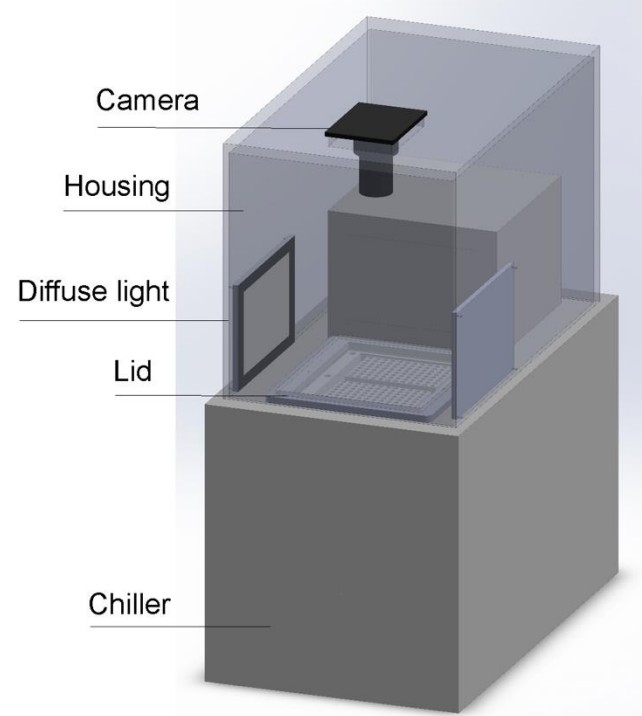
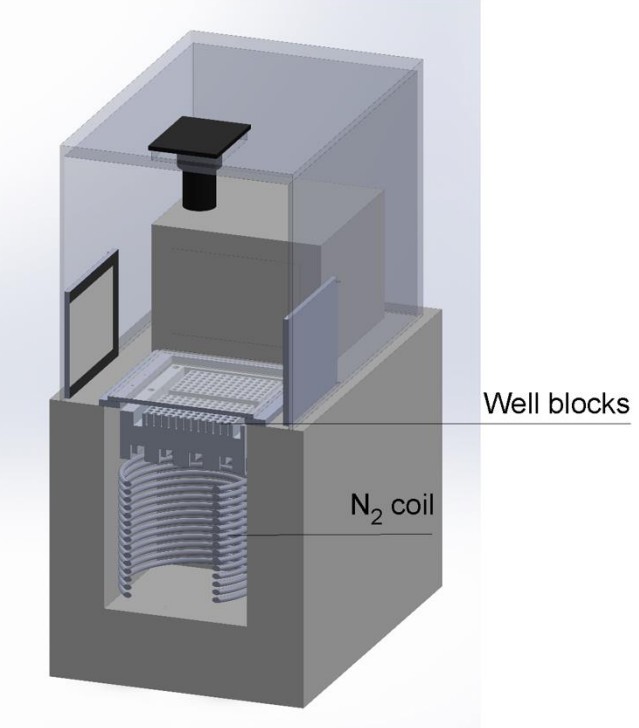

**Figure 1: Schematic of Automated Ice Spectrometer (AIS) showing primary components as indicated by labels.** Shown on the left is a camera, insulated housing for the camera and lights, LED diffuse lighting, a lid placed over the well blocks, and the chiller unit. Lid refers to the optically clear plexiglass cover, beneath which cold nitrogen gas is injected. Cutaway shows copper coil submerged in coolant that provides pre-chilled nitrogen gas to the space beneath plexiglass lid.



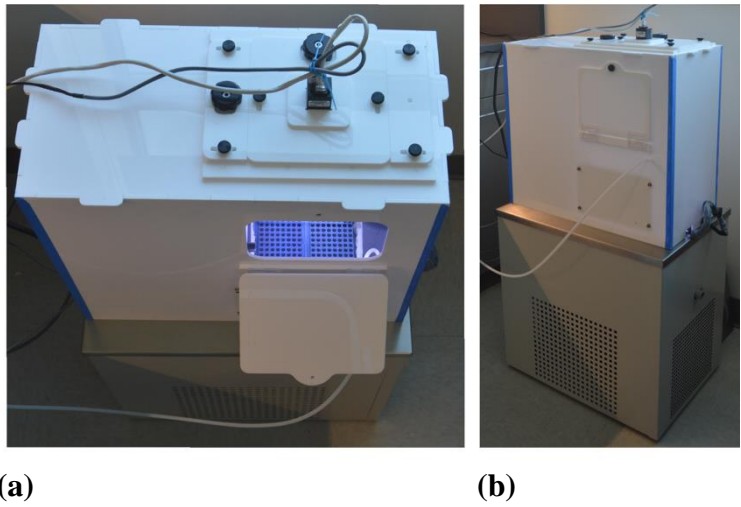

**(a)**                                    **(b)**

**Figure 2: Photo of current Automated Ice Spectrometer system.** Housing is manufactured from white-cast acrylic. Fig. 2a shows top-side view indicating camera and its adjustable cradle and access door open showing cooled well plate within. Fig. 2b oblique front view showing AIS housing sitting atop commercial chiller unit (Fisher Scientific Isotemp® Circulator). Tubing shown is the nitrogen gas input line.



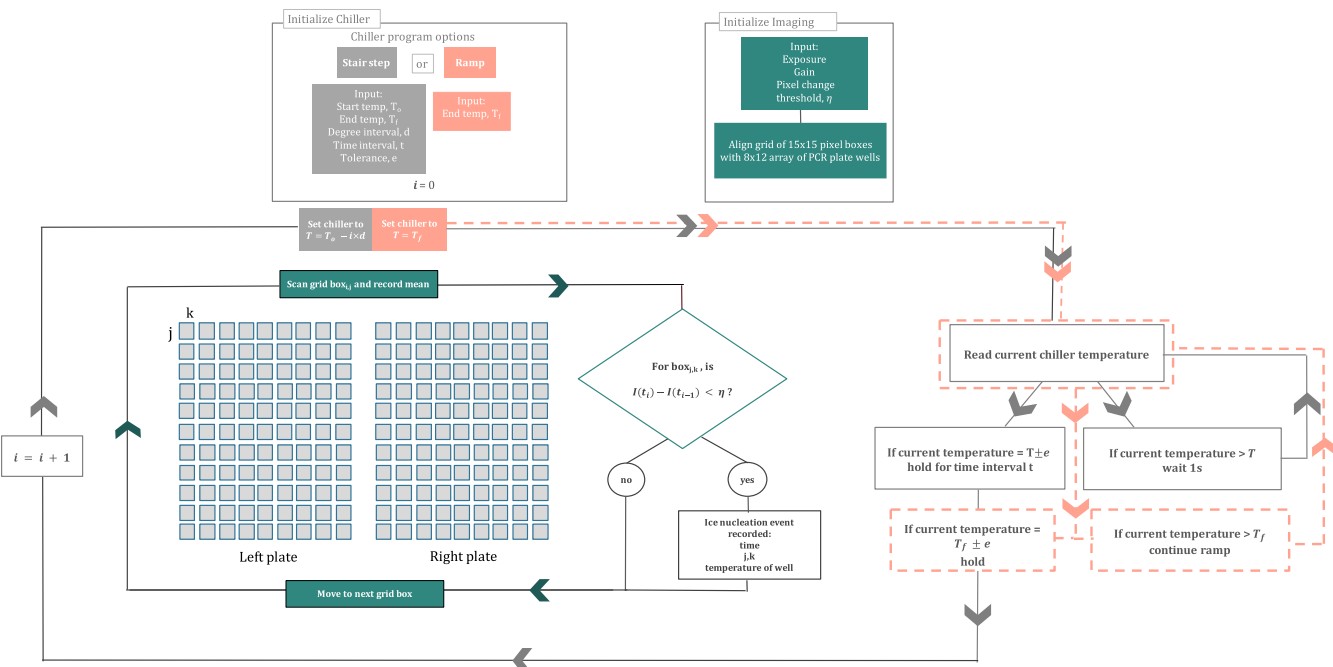

**Figure 3: Flow chart describing algorithm for detection of freezing events using camera, lights to leverage optical properties of phase change from water to ice.** Grey and pink used to indicate the "stair-step" and "ramp" temperature control of the chiller (for details see main text). Green used to indicate imaging section of the algorithm for detection of freezing events.



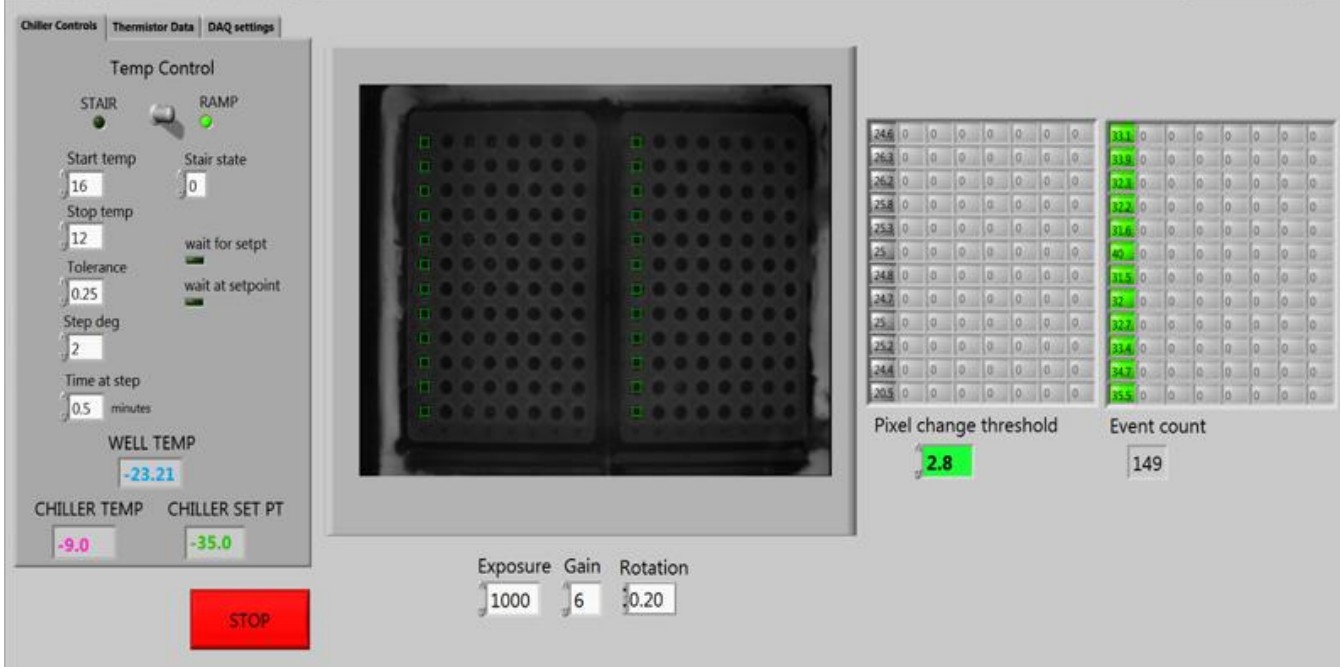

**Figure 4: A screenshot of the AIS user computer interface.** Chiller controls and well temperature readings on left, video stream of image of wells in middle. The detected freezing events are highlighted in green and displayed to the well matrix diagram on the right.





**Figure 5: Graph of well temperature vs. temperature offset measured between the base of the well and the air above the well as measured with a thermistor probe (orange filled circles).** The error bars in the larger plot show the range of temperature offsets corresponding to the ± 0.3°C calibration error of the temperature standard. From room temperature to -27 ℃ as measured at the base of the well, the average cooling rate is -0.86 ℃ min⁻¹. Inset shows cooling performance of the bath coolant, and the temperature within the well and gas above well over 6000 s measured by thermistor probes. The average cooling rate at the base of the well during the time period studied in the heat transfer simulation (from 0 ℃ to -27 ℃ min⁻¹, see Figs. 6, 7, and 8) however is -0.69 ℃ min⁻¹ because the cooling rate slows as the refrigerated cooling bath approaches its minimum temperature.



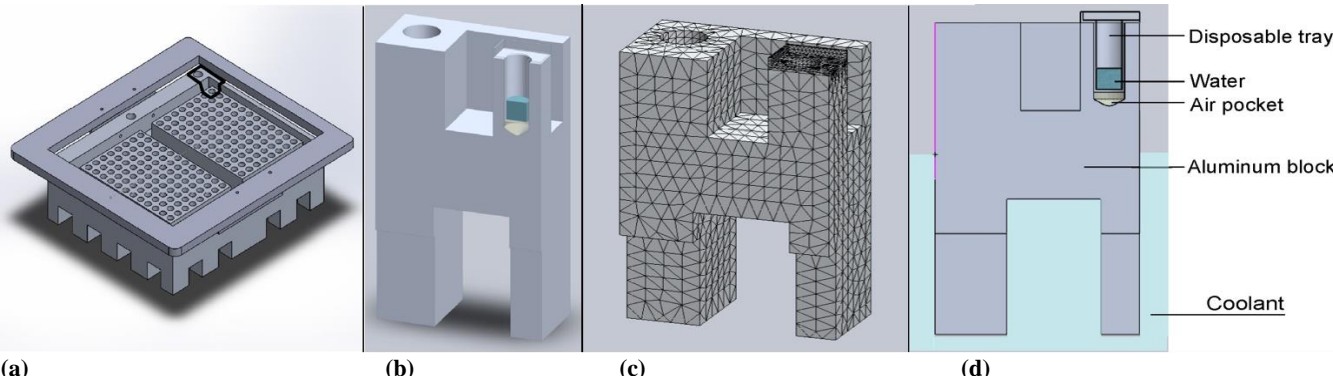

**(a)**          **(b)**          **(c)**          **(d)**

**Figure 6: Schematic of cut in well block made for heat transfer simulation and mesh applied.** (a) Well block and cover with black
5 line to indicate where cross-section (see Fig. 7b and 7c) in assembly was made for the heat transfer simulation. (b) Isometric, sectional
view of the block corner used in the heat transfer simulation. (c) Image of mesh applied using SolidWorks standard mesh solver (see Sec.
3.1 for details). (d) Isometric, flat sectional view of the block corner with labels to indicate materials parameterized in the heat transfer
simulations shown in Fig. 7 and Fig. 8.



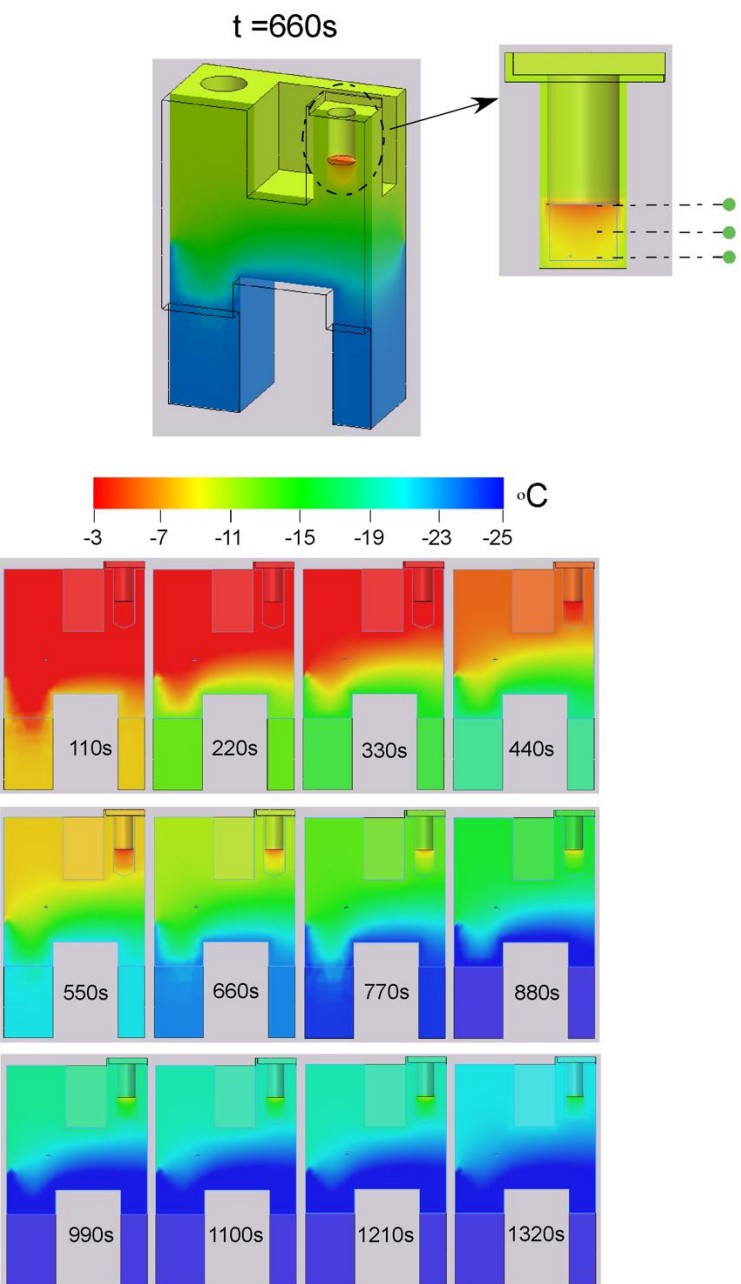

**Figure 7: Graphical time series of the heat transfer simulation. Top shows isometric view of the well block corner at t = 660 s.** Top right shows detailed plane view of well at t = 660 s. Dashed lines indicate temperature of water in the well at 3 points. Colors indicate temperature referenced by the scale below. Twelve time steps at 110 second intervals showing temperature distribution within the well block shown below. The average cooling rate over this specific time period is -0.69 °C min $^{-1}$. Results show the stratification of temperature in the well due to warmer air above the well region (air temperature fixed at +12 °C from the bath coolant temperature).

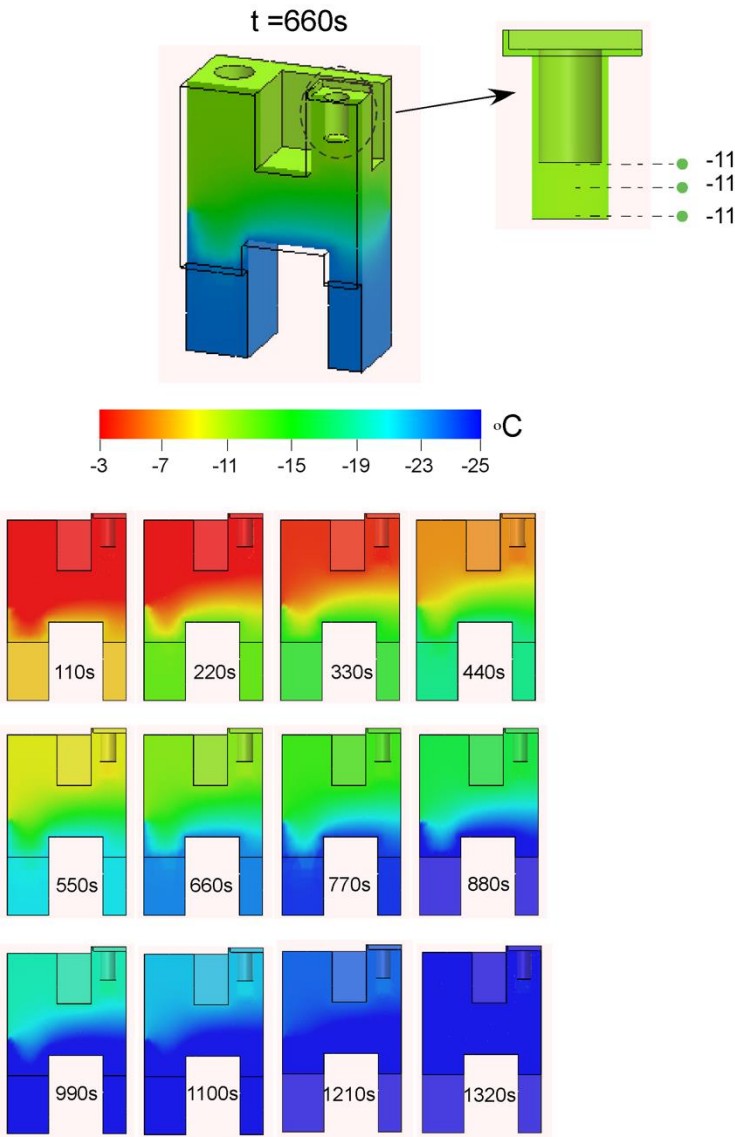

**Figure 8: Graphical time series of the heat transfer simulation.** The effects of increasing thermal homogeneity in the cooling
environment by efficiently cooling headspace gas on performance of the well block. As in Fig. 7, the top shows isometric view of the well
block corner at t = 660 s. Dashed lines indicate temperature of water in the well at 3 points. Colors indicate temperature referenced by the
scale below. Twelve time steps at 110 second intervals showing temperature distribution within the well block shown below. The average
cooling rate over this specific time period is -0.69 ℃ min$^{-1}$. Results show the lack of stratification of temperature in the well due to cooler
air above the well region (air temperature offset varied approximately +2-6 ℃ from the bath coolant temperature as shown in Fig. 5).







**Figure 9: Immersion freezing spectra of illite NX particles in terms of $n_{s,BET}$ (T) for comparison of SIO AIS against six other immersion mode techniques reported (see Hiranuma et al., 2015).** $n_{s,BET}$ is used to estimate ice nucleation surface-site density from an $N_2$ adsorption-based specific surface area (Hiranuma et al., 2015). CSU-IS (500 nm) represents measurements made on illite NX particles that were mobility diameter size-selected, whereas all other measurements reported were of bulk illite NX samples. Three different dilutions of illite NX suspensions were measured by the SIO AIS: 1:25,000 or $4.0 \times 10^{-5}$ wt %, 1:20,000 or $5.0 \times 5$ wt %, and 1:2,000 or $5.0 \times 10^{-4}$ wt %. SIO AIS measurements fall on the warm side of the spectra.