# Peer review of "Automation and Heat Transfer Characterization of Immersion Mode Spectroscopy for Analysis of Ice Nucleating Particles"

_Atmospheric Measurement Techniques, 2016_

## Referee Comment (RC1) · Anonymous Referee #2 · 28 Feb 2017

General Comments

The paper presents a modified immersion mode ice nucleation measurement technique. One improvement of the apparatus is automated control and analysis that enables for higher amounts of measurements leading to better statistics. But my opinion is that the more important part of the paper is the discussion of new heat transfer simulations supposed to get higher temperature accuracies in the development of such instruments. Especially this point is of great interest for the community in order to characterize and assess different instruments. However, the authors do not show the relationship between temperature measurement and sample temperature in detail. At the moment the description does not totally convince me that the new instrument is improved by the simulations. I have some specific comments and suggestions regarding the simulations and the literature comparison, and several technical comments.

Specific Comments

P4, L14: Additional reasons for high freezing temperatures might be impurities in the water that become important at µL volumes.

P5, L33: ... is smaller than ... (If the intensity decreases with freezing, $\eta$ should be a negative number (see also Fig. 4) or the opposite difference should be used: $I(t_{i-1}) - I(t_i) > \eta$)

P6, L1: $I(t_i) - I(t_{i-1}) < \eta$ (The correct equation is used in Fig. 3)

P6, L5: Please mark the position or hight of the thermistor in Fig. 6-8.

P8, L6: Where is something yellow in the inset of Fig. 5 (see also P9, L21)? I guess the black arrow shows the boundary conditions ("Data used in simulation"). This would mean a data range from about 800 s ($T_{well}$ = 0 °C) to 3000 s ($T_{well}$ = -27 °C). Why does the simulation cover only 1320 s of cooling?

P9, L20: Was the experimental bath temperature change with time used in the first simulation, too?

P9, L23: What does "a doubling of the warming mean"? The experimental temperature difference between the coolant bath and the headspace gas is about 10-11 °C after 800 s (below 0 °C).

P10, L11: ... +12 °C offset between the coolant bath and the headspace gas, ...

The paper would benefit from a graph which compares the simulation results. The interesting points for the reader would be the offset between the temperature of the water sample and the measured value at the well base, and the degree of stratification (the temperature gradient in the water sample).

P10, L15: The experimental temperature difference between the well and the headspace gas changes, and values from 3 to 5 °C are found in the well temperature region between 0 and -27 °C. The first simulation shows that stratification is present at 5.6 °C with an offset and a temperature gradient of about 3 °C. What are limit values for stratification and a temperature bias?

P10, L32: If the temperature offset should be quantified, why is that not a part of the current paper? Please comment on the difficulties of such a measurement here or add regarding results. I understand that the temperature measurement in the sample volumes does introduce a contamination, but a calibration using pure water or solutions with similar thermal properties should make a part of the interesting temperature range available. The authors do not use the simulation to quantify the offset. Why is the simulation not a reliable tool?

P11, L14: The dilution is a useless information without the starting concentration.

P11, L17: Why are no measurements done with lower concentrations? Reliable values should be possible down to freezing temperatures of -25 °C.

P11, L27: Comment on the reason why an offset to the data is not applied. Do probably show a plausible temperature range in Fig. 9 using error bars.

P12, L1: A quite high starting concentration of 10 mg/mL ($\sim$1 wt%) and a dilution of 1:25,000 would correspond to a mass concentration of $4 \cdot 10^{-4}$ mg/mL and a weight fraction of about $4 \cdot 10^{-5}$ wt%, assuming a density of 1 g/cm$^3$ and a negligible amount of solute. Please check values and units.

P12, L12: As the CSU-IS and the new SIO-AIS are almost identical focus on the comparison of these two instruments. The CSU-IS used 60 μL samples and weight fractions of 0.5 to $3 \cdot 10^{-6}$ wt% in the literature study. How can it be explained that the AIS data is in the high concentration range (low surface site densities and high freezing temperatures) whereas the concentrations are in the lower end?

P12, L12: To compare the AIS to all the other instruments in Fig. 9 additional data at lower concentrations and freezing temperatures between -20 and -25 °C are needed to ensure overlap. What is the reason for the difference to the other techniques? Can it be argued that the new one has a higher accuracy or are there other effects that have to be taken into account?

P13, L1: The experimental temperature differences are larger than 3 and 6 °C below a well temperature of -5 °C. Does this mean that stratification is present there?

P14, L1: See the comment on P12, L12.

P14, L6: The amount of samples per hour depends on the number of wells per sample. In principle the AIS can process 192 samples per cooling cycle. But to adequately characterize an IN it is necessary to investigate several concentrations and an amount of individual volumes larger than 24.

P14, L7: If the offsets are just identified but not quantified it does not lead to higher accuracies.

P16, Table 1: The Handbook of Chemistry and Physics gives thermal conductivities for air of 0.018 and 0.026 W m$^{-1}$ K$^{-1}$ at 200 and 300 K, respectively. Comment on the used values and add references.

To minimize a temperature bias between the aluminium block and the water sample the use of PCR wells made of thermally conductive plastics ($k$ = 5-10 W m$^{-1}$ K$^{-1}$) might be interesting.

P21, Fig. 5: Why is the well temperature higher than the headspace gas temperature at the beginning? Is the system in equilibrium at the start ($t$ = 0 s)?

P25, Fig. 9: There is less data for the CSU-IS (bulk) than presented by Hiranuma et al. (2015). Add the additional points if available.

There seems to be an amount of points that does not belong to one of the given instruments. Are these the missing CSU-IS values? Some points are similar but not identical. I suggest the use of different symbol types additionally to the colors and a legend for the attribution.

Technical Corrections

Page 1, Line 28: The AIS is compared to 6 other instruments in Fig. 9, not 5. Please correct that number whenever the comparison is discussed.

P2, L14: ... clouds. Accurately ...

P3, L18: ... the University of Colorado Raman Microscope Cold Stage (CU-RMCS) ...

P3, L24: ... frozen droplets ... grow by taking up water ... (the liquid droplets shrink)

P3, L31: ... change. Additionally, the heat transfer properties of the new SIO-AIS (Scripps Institution of Oceanography - Automated Ice Spectrometer) are characterized in the current paper through ...

P5, L9: ... temperature air away from the well ...

P5, L12: Change the figure numbers to the order of their appearance in the text: Fig. 5 → 3, and following.

P5, L17: ... National Instruments' LabVIEW software ...

P7, L10: ... $\pm$ 0.3 °C ...

P8, L6: ... is warmest in the simulation range, a maximum of 5.3 °C warmer ...

P8, L28: where $Q_{convection}$ is the rate ...

P9, L1: ... as two vertical plates, calculating the Nusselt number $N$, and using $h = Nk/H$, where H is ...

P9, L6: $Ra$ and $Pr$ are the Rayleigh and ...

P9, L8: $Ra = \frac{g\beta(T-T_\infty)D^3}{v^2} \cdot Pr$ (the viscosity has to be squared)

P9, L8: define the acceleration of gravity $g$

P9, L13: ... diffusivity. Since $\beta$,$v$, $\alpha$, and $k$ are ...

P11, L26: ... calculated using Eq. (1) (Vali, 1971). Although ...

P12, L1: ... The AIS measurement results in terms of cumulative INPs per volume were converted to the surface site density $n_{s,BET}$ using the mass concentration and the specific surface area as follows: ...

P12, L7: Add the BINARY here to get the 6 instruments of the comparison.

P12, L11: ... favourably to those of ...

P12, L26: ... from having to constantly monitor ...

P15, L27: Change the doi text color to black.

P15, L27: Insert a line break between the two articles.

P19, Fig. 3: Add the Tolerance $e$ to the Ramp option.

The text is still quite difficult to read in a printed version. Please use the font size of "Left plate" for the whole figure.

P21, Fig. 5: Clean the figure for ? symbols and use the same type of axis labels in all figures, e.g. "Temperature / °C".

Grids in both graphs similar to Fig. 9 would be beneficial here, too.

P21, L3: ... $\pm$ 0.3 °C ...

P21, L8: ... from 0 °C to -27 °C, see ...

---

## Referee Comment (RC2) · Anonymous Referee #1 · 6 Mar 2017

General comments: The work presented here is a valuable contribution to our understanding of droplet freezing measurements in ice nucleation research in general and a thorough description of a new Automated Ice Spectrometer (AIS) in particular. This instrument is an improved and automated version of the Colorado State University's Ice Spectrometer as it was described in the paper by Hiranuma et al. (2015). The manuscript presents the new instrument and its improvements against the old setup in detail. The novelty and significance of the work lie in the modelling of the thermal properties of the system. The results show that with the current setup the temperature of an aqueous immersion freezing sample in the AIS is homogeneous and that the freezing temperature can be measured accurately, provided the temperature probe is

placed properly. In a recent intercomparison of methods on NX Illite suspensions by Hiranuma et al., 2015 the immersion freezing temperatures were scattered by at least 5°C. The modelling results presented here may help to better understand which part of such observed discrepancies may come from inhomogeneities in sample temperature and from bias of temperature measurement. However, there are many other potential reasons for scattering between the different methods, such as sampling issues (filters vs. impingers) and differences in particle extraction from substrates. In the end the authors demonstrate that their improved AIS performs well by analyzing suspensions of the NX Illite test dust samples used by Hiranuma et al. (2015). At the warm end of ns vs. T spectra their data compare well to those of 5 other methods. Overall the manuscript is crafted well, and I recommend to publish it in ACP, after some of the points raised below have been addressed by the authors.

Specific scientific points:

P. 2, l. 2: I wonder what is meant by "the homogeneous freezing RH of aqueous solution droplet" ? Isn't the RH irrelevant, if you consider homogeneous freezing of liquid water ?

P. 5, L. 29: I do not understand why the intensity of light reflected back to the camera decreases when droplets freeze, as frozen drops become opaque and lighter than clear liquid drops, scattering more light towards the camera and leaving less for absorption at the dark background of the well block ? Please clarify.

P. 9, : A statement is required on how the equations (4) and (5) were derived, or where they are taken from.

Technical corrections:

Explain the acronym AIS upon first use: at present AIS occurs many times (e. g. in the abstract, and elsewhere) before it is first defined on page 5, L.1.

P. 2, L.13, insert "to" (identify drivers . . .)

P. 2, L. 24 and many other places: check the references in brackets: often names of authors are put in brackets, although the names are part of the sentence. An Example: “In (Hiranuma et al. , 2015), 17 online and offline . . . “

P. 3, L3: remove droplet assay

P.3, L. 3: Upon introducing the ice spectrometer the text refers to a paper by Hill et al. (2016) that is not in the reference list. Or do you mean Hill (2014)? Please clarify. P.3, L. 10: How can 50$\mu$l of water be filled into a 1.2$\mu$l well ? Please check the numbers.

P.3, L.19: introduce the acronym FRIDGE after “Frankfurt Ice Nuclei Deposition Freezing Experiment.”

P. 4, L. 23: Remove “Automated” in header of chapter 2.2

P. 5, L. 12: I presume that it’s not room air but heat that leaks into the nitrogen flow ? If so, then write something like “”. . . room air heat leaks into . . .”.

P.5, L. 32: "are" instead of "is" in: “time, freezing temperature, and location of the well are recorded . . .”

P. 8, L2: In Fig. 5 nothing is highlighted in yellow, as stated in line 2 and on P. 9, L. 16 . Please check.

Figure 5: In my copy the labels of all 4 axes as well as the legend and text in the insert have some strange characters (?). Please check.

References: Hiranuma et al.: A comprehensive laboratory study on the immersion freezing 45 behavior of illite NX particles: a comparison of seventeen ice nucleation measurement techniques, Atmos. Chem. Phys. Discuss., 14, accepted, doi:10.5194/acpd-14-22045-2014, 2015.

---

## Author Comment (AC1) · 11 May 2017

amt-2016-412 Automation and Heat Transfer Characterization of Immersion Mode Spectroscopy for Analysis of Ice Nucleating Particles Charlotte Beall, Dale Stokes, Thomas Hill, Paul DeMott, Jesse DeWald, and Kimberly Prather

We thank the reviewers for providing feedback on the manuscript and include our responses below.

Responses to Referee 2, Report 2

Specific Comments

P4, L14 "Additional reasons for high freezing temperatures might be impurities in the water that become important at $\mu$L volumes."

Thank you. L13-L15 now reads: "The homogenous freezing point of water is -38 C, but either the 96-well sample tray surface, or impurities present in the water induce freezing at higher temperatures, typically starting at -25 to -27 C, which limits the lower temperatures for which INP number concentrations may be assessed."

P5, L33: "... is smaller than ... (If the intensity decreases with freezing, $\eta$ should be a negative number (see also Fig. 4) or the opposite difference should be used: I(ti−1) − I(ti) > $\eta$)"

Whether the intensity increases or decreases with freezing depends on the color of the substrate, so we added absolute value bars. That way it applies whether the materials that show through the sample volumes are dark or light. The following change was made and reflected in both P5, L33 and Fig. 3 so that both are consistent with Fig. 4. |I(ti) − I(ti-1)| > $\eta$

P6, L1: "I(ti) − I(ti−1) < $\eta$ (The correct equation is used in Fig. 3)"

See response to P5, L33.

P6, L5: Please mark the position or hight of the thermistor in Fig. 6-8.

The height of the thermistor has been added to Fig. 6-8.

P8, L6: Where is something yellow in the inset of Fig. 5 (see also P9, L21)? I guess the black arrow shows the boundary conditions ("Data used in simulation"). This would mean a data range from about 800 s (Twell = 0 C) to 3000 s (Twell = -27 C). Why does the simulation cover only 1320 s of cooling?

Figure 5 (now Fig. 3) had been updated in previous versions of the manuscript with the same measurements to show a higher time resolution, but the simulation inputs had not been updated. I realize this could be confusing, particularly because as the referee

[Figure]

pointed out, the cooling rate was different. In order to address both this issue and the referee's suggestion (below) to quantify the offset between the well temperature and the thermistor, I made new measurements of the coolant and headspace gas temperature for the simulation input and remade Figures 5, 7-8. The new measurements were made to be sure that the calibration of the thermistor was up to date before using it to compare with the simulation output.

Originally, only 1320 seconds of cooling were shown in Fig. 7-8 in order to decrease the range of temperatures shown, and increase the resolution of the temperature gradient in the well block.

Fig. 5(3) has been updated to show the new measurements. Fig. 7-8 has been updated to show 3276 seconds of cooling, and a second colorbar was added to the close-up view of the well for higher resolution of the gradient.

P9, L20: "Was the experimental bath temperature change with time used in the first simulation, too?" Yes. P9, L20 now reads, "The simulation was run over 3276s with two different sets of boundary conditions representing the coolant fluid and headspace gas temperatures. In the first simulation, the coolant fluid temperatures from Fig. 3 were applied, but the difference between the temperature of the headspace gas and that of the well base was multiplied by two in order to approximate inefficient cooling of headspace gas."

P9, L23: What does "a doubling of the warming mean"? The experimental temperature difference between the coolant bath and the headspace gas is about 10-11 C after 800 s (below 0 C).

See response above to P9, L20.

P10, L11: ... +12 C offset between the coolant bath and the headspace gas, ... The paper would benefit from a graph which compares the simulation results. The interesting points for the reader would be the offset between the temperature of the water

sample and the measured value at the well base, and the degree of stratification (the temperature gradient in the water sample).

The requested figures have now been added as supplementary figures, Fig. S2 and Fig. S3.

P10, L15: The experimental temperature difference between the well and the headspace gas changes, and values from 3 to 5 C are found in the well temperature region between 0 and -27 C. The first simulation shows that stratification is present at 5.6 C with an offset and a temperature gradient of about 3 C. What are limit values for stratification and a temperature bias?

I think the limit values for stratification and temperature bias are defined by the user of the IN measurement, and depends on how precisely they want to measure IN activation temperatures. Depending on the buoyancy of the INP, degree and sign of the gradient, the activation temperature may differ from the measured temperature. Another issue to consider in the case of a cold to warm gradient from top to bottom of the sample volume. Although I am not aware of any publications on the effects of convection on ice nucleation, I imagine it is possible that convection could affect ice nucleation processes and ultimately the measured activation temperatures. I aim through the two simulations, to show that even under conditions in which the headspace gas is cooled considerably below room temperature, stratification can occur, and should be considered when reporting measurements. I think a 1°C threshold could be useful, because in (Hiranuma et al., 2015) instruments measuring wet suspensions of Illite agreed within 5°C (or 3 orders of magnitude in terms of ns). Perhaps a useful bias or stratification threshold should be at the same order of magnitude as the range of instrumental variability, decreasing as more agreement between instruments is achieved. In this manuscript however, I will consider stratification below the error associated with the thermal probe used as "unstratified".

P10, L12 now reads: "With an offset between the base of the well and the headspace

gas temperature of +3.0 °C, stratification has significantly decreased to 0.1 °C from top to bottom of the sample volume, which is within the error of the thermal probe (see Fig. S2)."

P10, L32: If the temperature offset should be quantified, why is that not a part of the current paper? Please comment on the difficulties of such a measurement here or add regarding results. I understand that the temperature measurement in the sample volumes does introduce a contamination, but a calibration using pure water or solutions with similar thermal properties should make a part of the interesting temperature range available. The authors do not use the simulation to quantify the offset. Why is the simulation not a reliable tool?

Originally, the output of the simulation had not been verified with measurements, so the simulation was not used quantitatively. Since the entire surface of the system was constrained by boundary conditions in the simulation, independent temperature measurements to check against the simulation must come from the interior of the well block. As shown in Fig. 5 (3), measurements from one thermistor placed underneath a well (and interior to the well block) were recorded simultaneously with the measurements of the coolant fluid and headspace gas, and these are independent from the boundary conditions. Thus, these measurements are used in Fig. S1 as a quantitative comparison against the simulated temperatures in the air pocket below the well at each of the twelve time steps, which enables the quantification of uncertainty.

In the original simulation, only the left upper corner well was included in the simulation in order to reduce computation time. However, I have since learned that symmetry is required for heat transfer models, so I redesigned the model to include the entire upper left quadrant of the well block. The simulation outputs featured in in Fig. S1-S4, and Fig. 7-8 now show the quadrant, and when compared with the thermistor embedded in the block in Fig. S1, are accurate within ± 0.6 °C. Now the simulation is a more reliable tool and can be used to quantify the offset between the sample volume and other regions of the well block.

It would also be useful to measure the freezing point of a liquid solution with a well characterized freezing temperature between 0 and -25°C, but I'm not aware of any solution with an appropriate homogenous freezing temperature. I think the same challenges motivated (Hiranuma et al., 2015), in their choice of Illite NX as a standard.

Changes to manuscript: Fig. 7-8, Fig. S1-S4.

P10 L32 now reads: "In order to verify the simulation output, simulated temperatures were checked against measurements that are independent of the simulation. Since the entire surface of the system was constrained by boundary conditions in the simulation, the measurements from inside of the well block at the well base (shown in Fig. 5) were used for comparison with the simulation output at the same location. Results of the comparison over the 12 time steps of the simulation are shown in Fig. S1. At subzero temperatures, the maximum difference between the measured and simulated temperatures was 0.6 °C at t = 819s, decreasing to values below the error of the thermistor for most of the simulation. 0.6 °C is assumed to be the uncertainty of the simulation. The measured temperature was consistently slightly warmer than the simulated temperature, possibly because the hole drilled into the aluminum well block was not modelled. In the second simulation using measured boundary conditions, the average temperature of the sample volume was compared with the average temperature of the air pocket in which the thermistor is placed throughout the 3276s simulation (see Fig. S3), in order to quantify the offset between the thermistor and the sample. The air pocket temperatures are consistently colder than the sample volume temperatures, ranging from -1.8 °C to -1.2 °C over the 3276s. Offsets in temperature between the 192 wells also exist in the AIS and are shown in Fig. S4. However, these results are not used quantitatively because there is currently only one thermistor embedded in the well block, so verification of the simulation's temperature gradient in x and y was not possible without further modifications to the well block. In the future, additional thermistors embedded within the well block can be used to verify the simulation output so that measurements can be adjusted with offsets due to the gradient in x and y as well

as z."

P11, L14: The dilution is a useless information without the starting concentration.

Thank you. The exact dilution procedure was added and corrected. P11 L14 now reads: "20 mg of illite NX was immersed in 500 mL of ultrapure water, resulting in a 4.0 $\times$ 10-3 wt % solution. Two more dilutions were made by immersing 25 mg of illite NX in 50mL of ultrapure water, and diluting again by factors of 1/10 and 1/100, resulting in solutions of 5.0 $\times$ 10-2 and 5.0 $\times$ 10-3 wt %, respectively. A final solution was prepared by starting with 300 mg in 50 mL of ultrapure water, then diluting by factors of 1/100 and 1/1000, resulting in a solution of 6.0 $\times$ 10-6 wt %." Fig. 9 was also updated to reflect these changes.

P11, L17: Why are no measurements done with lower concentrations? Reliable values should be possible down to freezing temperatures of -25 C.

A dilution of 6.0 x 10-6 wt % was measured and Fig. 9 was updated to show measurements down to -25 °C.

P11, L27: Comment on the reason why an offset to the data is not applied. Do probably show a plausible temperature range in Fig. 9 using error bars.

See response to P10, L32. Now that the output of the simulation has been compared against an independent measurement, I can quantify its uncertainty, as shown in Fig. S1. Error bars showing uncertainty due to temperature offset have been added to Fig. 9.

P12, L1: A quite high starting concentration of 10 mg/mL (âĹij1 wt%) and a dilution of 1:25,000 would correspond to a mass concentration of 4Âů10−4 mg/mL and a weight fraction of about 4Âů10−5 wt%, assuming a density of 1 g/cm3 and a negligible amount of solute. Please check values and units.

See comment for P11, L14. Thank you.

P12, L12 (a): As the CSU-IS and the new SIO-AIS are almost identical focus on the comparison of these two instruments. The CSU-IS used 60 $\mu$L samples and weight fractions of 0.5 to 3Âů10−6 wt% in the literature study. How can it be explained that the AIS data is in the high concentration range (low surface site densities and high freezing temperatures) whereas the concentrations are in the lower end?

Now that the concentrations have been corrected, we know that the dilutions fall in the middle of the range rather than the low end. AIS data is still in the upper end of the spread between the 6 instruments featured in Fig. 9.

The heat transfer simulations demonstrate how sensitive temperature measurements are to the placements of the probe and the thermal homogeneity of the environment. To my knowledge, the locations of thermal probes are not reported for the instruments featured in Fig. 9, so it is difficult to speculate on potential sources of bias for the other instruments. However, if there are similarities in the various instruments' thermal properties, it is possible that the higher heat content of the sample volume relative to the rest of the system is consistent across multiple instruments. If this is true, since thermal probes cannot be in contact with the sample volume directly for the reasons described in Sec. 3.1, it is possible that they are in thermal contact with regions of the instruments which are colder than the sample volumes. For example, a thermal probe could be placed in the hole that is seen in the left top corner of Fig 6 b. A probe inserted into this hole could result in cold temperature bias, because the temperature of the air along the z axis of this hole is colder than the well temperature. So I believe that the AIS data might fall on the high concentration side, because the thermistor is so close and in thermal contact (with heat sink compound) with the sample volume.

P12, L12 (b): To compare the AIS to all the other instruments in Fig. 9 additional data at lower concentrations and freezing temperatures between -20 and -25 C are needed to ensure overlap. What is the reason for the difference to the other techniques? Can it be argued that the new one has a higher accuracy or are there other effects that have to be taken into account?

Additional data at lower concentrations has been added to Fig. 9, and freezing temperature are consistently higher than the other instruments. Barring any inconsistency in the standard Illite NX sample itself, potential cold biases are possible and discussed above in the response to P12, L12 (a). Now that the simulation is used quantitatively, I believe it is possible to achieve higher accuracies with the AIS in comparison with the other instruments because sources of bias in freezing temperature measurements can be identified, quantified and mitigated. For example, Fig. S4 shows the gradient in sample volume temperatures from left to right over a cross section of the well block. Wells closer to the outside of the system are up to 2.2 °C warmer than wells closer to the interior of the block. Eventually I plan to use the simulation to map out the spatial temperature gradient in x, y and z, to determine offsets for each well from the thermistor embedded in the well block. However, as I would like to verify the simulation output in more than one location before I determine the offsets in x-y and z for all 192 wells, I represent the maximum offsets found through the simulations in Fig. 9 with error bars in this paper. In the z-direction, the offset between the thermistor embedded in the well block and the warmer sample volume reaches a maximum of -1.8 °C at subzero temperatures (see Fig. S2). In the x and y direction, the difference between colder sample volumes near the interior of the block and warmer sample volumes near the outer perimeter is a maximum of +2.2 °C. In summary, the z-direction temperature gradient presents a likely cold bias to the measurements, while the x and y temperature gradients present an opposing warm bias, that is greater than the cold bias. So the greater of the two uncertainties, ±2.2 °C, is represented in Fig. 9 with error bars.

P14, L16 now reads: "The heat transfer simulations applied here could support investigations of bias in temperature measurement for INP measurement techniques, enable higher accuracy in INP freezing temperature measurements, and ultimately help decrease disparities between various instruments."

P13, L1: The experimental temperature differences are larger than 3 and 6 åŮęC below a well temperature of -5 åŮęC. Does this mean that stratification is present there?

There is indeed stratification present within the well in both simulations, as shown in the close-up of the sample volume in Fig. 7 and Fig. 8. The degree of stratification present in both simulations is shown in Fig. S2, reaching a maximum of +0.6 °C in Simulation 1 and +0.2 °C in Simulation 2. P10, L24 reads: "With an offset between the base of the well and the headspace gas temperature of +3.0 °C, stratification has significantly decreased to 0.1 °C from top to bottom of the sample volume, which is within the error of the thermal probe (see Fig. S2)."

P14, L1: See the comment on P12, L12.

Thank you, a more dilute solution of Illite NX was used in order to cover the freezing temperatures from -20 °C to -25 °C.

P14, L6: The amount of samples per hour depends on the number of wells per sample. In principle the AIS can process 192 samples per cooling cycle. But to adequately characterize an IN it is necessary to investigate several concentrations and an amount of individual volumes larger than 24.

P14, L6 now reads: "...can process up to 7 samples per hour using 24 wells per sample, over 4 times faster throughput than older versions of the instrument (including time for loading samples )".

P14, L7: If the offsets are just identified but not quantified it does not lead to higher accuracies. See response to P12, L12(b).

P16, Table 1: The Handbook of Chemistry and Physics gives thermal conductivities for air of 0.018 and 0.026 W m$-1$ K$-1$ at 200 and 300 K, respectively. Comment on the used values and add references.

Thank you. 0.027 W m-1 K-1 was used in the simulation and Table 1 has been corrected with references added.

To minimize a temperature bias between the aluminium block and the water sample the use of PCR wells made of thermally conductive plastics (k = 5-10 W m$-1$ K$-1$) might

be interesting.

True, thank you. The trouble would be finding one with a smooth consistent surface.

P21, Fig. 5: Why is the well temperature higher than the headspace gas temperature at the beginning? Is the system in equilibrium at the start (t = 0 s)?

Because the thermistor is calibrated for the range 5 °C to -35 °C, the thermistor does not accurately measure temperatures at higher temperatures. So the well temperature is not necessarily warmer at t = 0s.

P25, Fig. 9: There is less data for the CSU-IS (bulk) than presented by Hiranuma et al. (2015). Add the additional points if available.

There seems to be an amount of points that does not belong to one of the given instruments. Are these the missing CSU-IS values? Some points are similar but not identical. I suggest the use of different symbol types additionally to the colors and a legend for the attribution.

I checked the figure again, and I see all 6 measurements there and labeled (7 including both CSU-IS measurements).

Technical corrections

Page 1, Line 28: The AIS is compared to 6 other instruments in Fig. 9, not 5. Please correct that number whenever the comparison is discussed.

Corrected, thank you.

P2, L14: ... clouds. Accurately ...

The strikethrough was removed.

P3, L18: ... the University of Colorado Raman Microscope Cold Stage (CU-RMCS) ...

L18 now reads: ". . .the University of Colorado Raman Microscope Cold Stage (CU-RMCS) (Baustian et al., 2010; Wise et al. 2010). . ."

P3, L24: ... frozen droplets ... grow by taking up water ... (the liquid droplets shrink)
L24 now reads: ". . .in which frozen droplets near liquid droplets take up water vapour as the liquid droplets shrink. . ."

P3, L31: ... change. Additionally, the heat transfer properties of the new SIO-AIS (Scripps Institution of Oceanography - Automated Ice Spectrometer) are characterized in the current paper through ...

L31 now reads: Additionally, the heat transfer properties of the new SIO-AIS (Scripps Institution of Oceanography – Automated Ice Spectrometer) instrument are also characterized through.."

P5, L9: ... temperature air away from the well ...

L9 now reads: "The cold nitrogen gas purges room temperature air away from of the well region".

P5, L12: Change the figure numbers to the order of their appearance in the text: Fig. 5 → 3, and following.

Fig. 5 is now Fig. 3, and the old Fig. 3 and Fig 4 have shifted down one number accordingly.

P5, L17: ... National Instruments' LabVIEW software ...

VIEW has been capitalized.

P7, L10: ... $\pm$ 0.3 C ...

Spacing has been added.

P8, L6: ... is warmest in the simulation range, a maximum of 5.3 C warmer ...

L6 now reads: "Figure 5 shows that the air above the well region is warmest in the simulation range, a maximum of +5.28 °C."

P8, L28: where Qconvection is the rate ...

"The" has been added to this phrase in L28.

P9, L1: ... as two vertical plates, calculating the Nusselt number N, and using h = Nk/H, where H is ...

P9, L1 now reads: "...estimated by approximating the wells as two vertical plates, calculating the Nusselt number N, and using h=Nk/H, where..."

P9, L6: Ra and P r are the Rayleigh and ...

Spelling of Rayleigh number was corrected.

P9, L8: Ra... Pr (the viscosity has to be squared)

Corrected the equation by adding the square, thank you.

P9, L8: define the acceleration of gravity g

Added a definition of g in L8.

P9, L13: ... diffusivity. Since $\beta, v, \alpha$, and k are ..

Italicized k in L13.

P11, L26: ... calculated using Eq. (1) (Vali, 1971). Although ...

P11, L26 now reads: "Cumulative concentration of INPs per volume per 0.25 °C were calculated using Eq. (1) (Vali, 1971)."

P12, L1: ... The AIS measurement results in terms of cumulative INPs per volume were converted to the surface site density ns,BET using the mass concentration and the specific surface area as follows: ...

P12, L1 verbiage altered to reflect above.

P12, L7: Add the BINARY here to get the 6 instruments of the comparison.

Added BINARY to the list of instruments in the comparison.

P12, L11: ... favourably to those of ...

P12, L11 now reads: "...the Automated Ice Spectrometer measurements compare favourably to those of the other 6 techniques..."

P12, L26: ... from having to constantly monitor ...

P12, L26 now reads: "...and frees the operator from having to constantly monitor sample processing."

P15, L27: Change the doi text color to black.

Text changed to black.

P15, L27: Insert a line break between the two articles.

Corrected spacing, thank you.

P19, Fig. 3: Add the Tolerance e to the Ramp option.

Fig. 3: Tolerance e was added to the Ramp option.

P21, Fig. 5: Clean the figure for ? symbols and use the same type of axis labels in all figures, e.g. "Temperature / C". Fig. 5, strange characters were eliminated.

Grids in both graphs similar to Fig. 9 would be beneficial here, too. Fig. 5: both grids added.

P21, L3: ... $\pm$ 0.3 C ... P21, L3, space added.

P21, L8: ... from 0 C to -27 C, see ... P21, L8 has been changed due to the new measurements.

Please also note the supplement to this comment:
http://www.atmos-meas-tech-discuss.net/amt-2016-412/amt-2016-412-AC1-supplement.pdf

[Figure]

Camera

Housing

Diffuse light

Lid

Chiller

Well blocks

N$_2$ coil

**Fig. 1.** Schematic of Automated Ice Spectrometer (AIS) showing primary components as indicated by labels.

[Figure]

**Fig. 2.** Photo of current Automated Ice Spectrometer system.

[Figure]

**Fig. 3.** Graph of well temperature vs. temperature offset measured between the base of the well and the air above the well as measured with a thermistor probe (orange filled circles).

[Figure]

**Fig. 4.** Flow chart describing algorithm for detection of freezing events using camera, lights to leverage optical properties of phase change from water to ice.

[Figure]

**Fig. 5.** A screenshot of the AIS user computer interface.

[Figure]

Disposable tray
Water
Air pocket/Thermistor

Aluminum block

Coolant

**Fig. 6.** Schematic of cut in well block made for heat transfer simulation and mesh applied.

**Fig. 7.** Graphical time series of the heat transfer simulation. Top shows isometric view of the well block (top left quarter) at t = 1638 s.

**Fig. 8.** Graphical time series of the heat transfer simulation.

[Figure]

**Fig. 9.** Immersion freezing spectra of illite NX particles in terms of ns,BET (T) for comparison of SIO AIS against six other immersion mode techniques reported (see Hiranuma et al., 2015).

---

## Author Comment (AC2) · 11 May 2017

amt-2016-412 Automation and Heat Transfer Characterization of Immersion Mode Spectroscopy for Analysis of Ice Nucleating Particles Charlotte Beall, Dale Stokes, Thomas Hill, Paul DeMott, Jesse DeWald, and Kimberly Prather

We thank the reviewers for providing feedback on the manuscript and include our responses below.

Responses to Referee 1, Report 1

P. 2, l. 2: I wonder what is meant by "the homogeneous freezing RH of aqueous solution droplet" ? Isn't the RH irrelevant, if you consider homogeneous freezing of

liquid water ?

Yes, thank you, this is true for immersion mode ice nucleation, but we intend to define INPs in general before narrowing the discussion to immersion mode INPs only. In deposition ice nucleation studies, RH is an important factor.

P. 5, L. 29: I do not understand why the intensity of light reflected back to the camera decreases when droplets freeze, as frozen drops become opaque and lighter than clear liquid drops, scattering more light towards the camera and leaving less for absorption at the dark background of the well block ? Please clarify

Whether the intensity increases or decreases with freezing depends on the color of the substrate, so we added absolute value bars. That way it applies whether the materials that show through the sample volumes are dark or light.

The following change was made and reflected in both P5, L33 and Fig. 3 so that both are consistent with Fig. 4. $|I(t_i) - I(t_{i-1})| > \eta$

P. 9, : A statement is required on how the equations (4) and (5) were derived, or where they are taken from.

A citation for equation (4) was added: Churchill and Chu, 1975.

Technical corrections: Explain the acronym AIS upon first use: at present AIS occurs many times (e. g. in the abstract, and elsewhere) before it is first defined on page 5 L.1

Thank you. AIS is defined in the abstract upon first use now, and on P3 L32.

P. 2, L.13, insert "to" (identify drivers . . .)

P.2, L.13 now reads "..are needed to identify drivers. . ."

P. 2, L. 24 and many other places: check the references in brackets: often names of authors are put in brackets, although the names are part of the sentence. An Example:

"In (Hiranuma et al. , 2015), 17 online and offline . . . "

It appears that this bracket is standard formatting in AMT? From the website: "In general, in-text citations can be displayed as "[. . .] Smith (2009) [. . .]", or "[. . .] (Smith, 2009) [. . .]"."

http://www.atmospheric-measurement-techniques.net/for_authors/manuscript_preparation.html

P. 3, L3: remove droplet assay

Thank you. P.3, L3 "droplet assay" has been removed.

P.3, L. 3: Upon introducing the ice spectrometer the text refers to a paper by Hill et al. (2016) that is not in the reference list. Or do you mean Hill (2014)?

Yes, correct. The reference should read Hill et al., 2014. P.3, L.3 now reads (Hill et al., 2014).

Please clarify. P.3, L. 10: How can $50\mu l$ of water be filled into a $1.2\mu l$ well ? Please check the numbers.

P.3, L.10 corrected and now reads ". . .small aliquots of water, typically around 50 $\mu$L each, are distributed in 1.2 mL wells. . ."

P.3, L.19: introduce the acronym FRIDGE after "Frankfurt Ice Nuclei Deposition Freezing Experiment."

P.3, L.19 now reads ". . .the Frankfurt Ice Nuclei Deposition FreezinG Experiment (FRIDGE). . ."

P. 4, L. 23: Remove "Automated" in header of chapter 2.2

The header of chapter 2.2 is now "Physical Design of the Ice Spectrometer."

P. 5, L. 12: I presume that it's not room air but heat that leaks into the nitrogen flow ? If so, then write something like "". . . room air heat leaks into . . .".

Yes, actually, if flow rates are too low, we believe room air does leak into the system because it is not perfectly insulated. The acrylic lid for example, is not sealed, so if the flow of nitrogen is low, room air could seep in underneath the lid.

P.5, L. 32: "are" instead of "is" in: "time, freezing temperature, and location of the well are recorded . . ."

P.5, L. 32: now reads ". . .time, freezing temperature, and location of the well are recorded."

P. 8, L2: In Fig. 5 nothing is highlighted in yellow, as stated in line 2 and on P. 9, L. 16 . Please check.

Thank you, the measurements were updated in response to the other reviewer, so there is no longer a yellow highlighted feature in the manuscript.

Figure 5: In my copy the labels of all 4 axes as well as the legend and text in the insert have some strange characters (?). Please check.

This is another problem due to the pdf conversion. I will make sure that the characters convert correctly.

Reference:

Thank you, the updated reference of (Hiranuma et al., 2015) was added.

Please also note the supplement to this comment:
http://www.atmos-meas-tech-discuss.net/amt-2016-412/amt-2016-412-AC2-supplement.pdf

Camera

Housing

Diffuse light

Lid

Chiller

Well blocks

$N_2$ coil

**Fig. 1.** Schematic of Automated Ice Spectrometer (AIS) showing primary components as indicated by labels.

[Figure]

**Fig. 2.** Photo of current Automated Ice Spectrometer system

[Figure]

Fig. 3. Graph of well temperature vs. temperature offset measured between the base of the
well and the air above the well as measured with a thermistor probe (orange filled circles).

[Figure]

**Fig. 4.** Flow chart describing algorithm for detection of freezing events using camera, lights to leverage optical properties of phase change from water to ice.

[Figure]

**Fig. 5.** A screenshot of the AIS user computer interface.

[Figure]

**Fig. 6.** Schematic of cut in well block made for heat transfer simulation and mesh applied.

[Figure]

Fig. 7. Graphical time series of the heat transfer simulation. Top shows isometric view of the well block (top left quarter) at t = 1638 s.

[Figure]

**Fig. 8.** Graphical time series of the heat transfer simulation.

Fig. 9. Immersion freezing spectra of illite NX particles in terms of ns,BET (T) for comparison
of SIO AIS against six other immersion mode techniques reported (see Hiranuma et al., 2015).

In the figure: NC State-CS; CU-RMCS; FRIDGE (imm. mode); SIO AIS 4.0 × 10⁻³ weight %; SIO AIS 5.0 × 10⁻³ weight %; SIO AIS 5.0 × 10⁻² weight %; SIO AIS 6.0 x 10⁻⁶ weight %; BINARY; CSU-IS (500-nm); Leeds-NIPI; CSU-IS (bulk); axes Temperature (°C) and $n_{s,\mathrm{BET}},\ m^{-2}$.

**Supplement:**

[Figure]

**Figure S1: Difference between measured and simulated temperature of the air pocket in the well base.** The maximum difference between the predicted temperature and the measured temperature over the twelve time steps of the 3276s simulation was 0.6 ºC at subzero temperatures. The error bars shown represent the uncertainty of the thermal probe at the well base, ±0.3 ºC.

[Figure]

**Figure S2: Simulated temperature stratification within the sample volume.** The difference in simulated temperature between the top and bottom of the 50 $\mu L$ sample volume is shown for the two simulations. Stratification of the sample volume under normal conditions as measured in the AIS reaches a maximum of 0.2 ºC. Under the warmer headspace gas conditions as described in Sec. 3.2, stratification increases to a maximum of 0.6 ºC.

[Figure]

**Figure S3: Simulated temperature offset between the sample volume and the well base.** The average sample volume temperature is warmer than the air pocket in the well base throughout the 3276s simulation. Error bars reflect the uncertainty of the simulation based on the difference between measured and simulated temperature of the well base (see Fig. S1).

[Figure]

**Figure S4: Spatial temperature gradient in sample volume temperature.** Cross-section of top left quadrant of the well block (see Sec. 3.2 for details) at 1638 s. The sample volumes are outlined by the thick black rectangles. The leftmost sample volumes are closer to the outer edges of the well block, whereas the rightmost sample volumes are closer to the interior of the block. Thus, the leftmost sample volumes that are closer to the outer edges of the well block are warmer than the sample volumes along the interior of the well block, with a maximum difference of 2.2°C.